# A Bearing Fault Classification Framework Based on Image Encoding Techniques and a Convolutional Neural Network under Different Operating Conditions

**DOI:** 10.3390/s22134881

**Published:** 2022-06-28

**Authors:** Rafia Nishat Toma, Farzin Piltan, Kichang Im, Dongkoo Shon, Tae Hyun Yoon, Dae-Seung Yoo, Jong-Myon Kim

**Affiliations:** 1Department of Electrical, Electronics and Computer Engineering, University of Ulsan, Ulsan 44610, Korea; rafiatoma.eceku@gmail.com (R.N.T.); piltanfarzin@gmail.com (F.P.); 2ICT Convergence Safety Research Center, University of Ulsan, Ulsan 44610, Korea; kichang@ulsan.ac.kr; 3Electronics and Telecommunications Research Institute (ETRI), Daejeon 34129, Korea; sdk@etri.re.kr (D.S.); thyoon0820@etri.re.kr (T.H.Y.); ooseyds@etri.re.kr (D.-S.Y.)

**Keywords:** bearing fault diagnosis, convolutional neural network (CNN), gramian angular field (GAF), motor-current signal, time-series imaging

## Abstract

Diagnostics of mechanical problems in manufacturing systems are essential to maintaining safety and minimizing expenditures. In this study, an intelligent fault classification model that combines a signal-to-image encoding technique and a convolution neural network (CNN) with the motor-current signal is proposed to classify bearing faults. In the beginning, we split the dataset into four parts, considering the operating conditions. Then, the original signal is segmented into multiple samples, and we apply the Gramian angular field (GAF) algorithm on each sample to generate two-dimensional (2-D) images, which also converts the time-series signals into polar coordinates. The image conversion technique eliminates the requirement of manual feature extraction and creates a distinct pattern for individual fault signatures. Finally, the resultant image dataset is used to design and train a 2-layer deep CNN model that can extract high-level features from multiple images to classify fault conditions. For all the experiments that were conducted on different operating conditions, the proposed method shows a high classification accuracy of more than 99% and proves that the GAF can efficiently preserve the fault characteristics from the current signal. Three built-in CNN structures were also applied to classify the images, but the simple structure of a 2-layer CNN proved to be sufficient in terms of classification results and computational time. Finally, we compare the experimental results from the proposed diagnostic framework with some state-of-the-art diagnostic techniques and previously published works to validate its superiority under inconsistent working conditions. The results verify that the proposed method based on motor-current signal analysis is a good approach for bearing fault classification in terms of classification accuracy and other evaluation parameters.

## 1. Introduction

Induction motors (IM) are one of the critical components in modern manufacturing industries for maintaining production chains. They offer a simple control mechanism, minimal cost requirements, a robust design, and great reliability. Induction motors are so pervasively used that almost 40% of the world’s electric power is consumed by this category [1]. The application fields of IM do not only belong to the electric utility and mining industries, but also to aerospace, nuclear plants, and military activities [2]. Distinct types of faults could occur because of the continuous electromechanical stresses in the harsh and severe working surroundings, and initiations of these faults can show subtle or tangible changes in the physical signals that are used for condition monitoring. Gradually, the initial symptoms could result in the sudden failure of rotating components and cause personnel causalities, as well as high economic losses [3]. Sudden failures also lead to increased downtime, which could cost more than the actual cost of the motor. If the faults can be identified in the initial state of the condition-monitoring process, downtime and financial loss can be lessened [4]. To avoid such unexpected situations, research on fault-detection techniques has become a very delicate issue for researchers over the past few years. Motor faults can be classified based on the components, such as bearing faults (40%), rotor faults (10%), stator faults (38%), and others (12%). This indicates that faults that occur due to bearings are the most common in practical scenarios [5,6]. In general, the fault-diagnosis methods can be separated into model-based, signal-based, data-driven, and hybrid approaches. A suitable amount of historical data is necessary to implement the data-driven approach which will help to differentiate the fault modes of the system. A good amount of sensor data is now available for condition-monitoring systems due to the huge development of sensor technologies which represent the various states and fault types of IM. The availability of sensor data is helping researchers to build robust data-driven fault diagnosis models and reduce the possibility of faults causing sudden failures in the industry [7]. Different types of signals captured by sensors, such as vibration [8], current [9,10], temperature [11], acoustic emission [12], and stray flux [13] are used in data-driven techniques. Identifying the technical conditions of an IM by analyzing the extracted damage features from the measured signal is one of the main ideas behind the data-driven methods.

The vibration sensors, i.e., accelerometers, are used around the bearing in the testbed to acquire the vibration signals in different bearing conditions, which reflect the status of the health condition of the bearing. However, the overall system installation process is complex, the cost is quite high, and the system needs full-time monitoring to observe the bearing conditions [14]. Due to this, vibration-based condition monitoring is not always easy to implement in remote places. In recent times, the current signal-based data-driven techniques have attracted researchers as they do not need any external sensors for collecting data. However, with only current transducers, the single-phase stator current can be recorded, where no extra current transformers or frequency inverters are required [15]. This low-cost and non-invasive technique of data collection makes motor-current signal analysis (MCSA) a suitable mechanism for condition monitoring and fault diagnosis. Further, the sensor fusion scheme that integrates both the vibration and current signals were also investigated to improve the fault-classification accuracy of the induction motor [16].

Most of the time, the collected signals do not exhibit distinct fault signatures due to the surrounding noise. Therefore, different signal-processing techniques, such as time-domain, frequency-domain, or time–frequency domain techniques are employed to find the fault features [17]. Time-domain signals exhibit the basic representation of data, whereas frequency-domain signals can separate the characteristics of the fault frequencies because of their high processing gain and low noise sensitivity. In some cases, frequency-domain analysis does not perform well in non-stationary signals. For such cases, time–frequency domain approaches are used, which can extract fault characteristics from both stationary and non-stationary signals. However, it becomes very difficult to extract accurate fault features from these signal processing techniques due to the variable working conditions and noisy environment around the IM, which makes research on the condition monitoring of IMs trickier and more challenging for researchers [3].

In recent times, the combination of signal processing and machine-learning (ML) algorithms has been utilized in the field of rolling-element bearing fault diagnosis [18,19,20]. In general, after the data-acquisition step, statistical features from the time domain, such as root mean square (RMS), kurtosis, skewness, and frequency-domain features originating from Fourier transforms are mostly used as the input of ML classifiers. Hence, it is not guaranteed that these features can effectively distinguish among multiple health conditions of the bearing due to the non-linear and non-stationary characteristics of the acquired signals. In some cases, a time–frequency domain analysis is used to decompose the signals into a series of components that contain both time- and frequency-domain information. Commonly applied time–frequency methods include wavelet transform; short-time Fourier transform; Wigner–Ville distribution (WVD); empirical mode decomposition (EMD); and enhanced empirical-mode decomposition (EEMD) [21]. The next step is to apply ML algorithms to the extracted features to classify different states of the bearing, where the algorithm provides the theoretical base for designing an efficient model. Support vector machines (SVM) [22,23,24]; k-nearest neighbors (KNN) [20]; and artificial neural networks (ANN) [25,26] are found to be effective ML algorithms in fault diagnosis tasks. Hence, profound skill and expertise in signal processing techniques are required to effectively run the feature extraction and selection method to achieve high efficiency in the classification process. Further, any manual feature-selection method for any certain task may not be effective in different working scenarios. Another fact to consider is that the traditional ML algorithms cannot extract features automatically from the raw data and always need an efficient feature extraction process to provide good results, which is not only time-consuming but also requires more manpower [27,28]. Real-time fault identification also becomes important to lessen the probability of sudden failure and ensure long-time operation to improve the industrial environment. To ensure this, the online monitoring of the faulty conditions of IM was also investigated in [29] by combining the Internet of Things (IoT) architecture and effective ML techniques to recognize and classify the faulty conditions immediately.

The deep learning (DL)-based fault diagnosis method can be a helpful solution to the drawbacks of traditional data-driven fault diagnosis methods based on ML algorithms. Due to the automatic feature-extraction capabilities from the original signals, the DL-based fault diagnosis becomes a promising tool in fault diagnosis. In the DL-based method, multi-scale feature extraction and the final pattern recognition can be achieved altogether through stacking multiple layers in a detailed hierarchical architecture [30]. Generally, collecting time-series data from sensors and directly providing them as the input of the DL model to obtain the fault-classification results for multiple faults are the basic steps of the DL-based fault-diagnosis approach. However, in recent times, researchers extract features first and then apply them to the DL model to obtain high-level features to automatically classify the faults and improve model accuracy [31]. The DL-based fault diagnosis mainly consists of deep auto-encoders [32,33], a deep belief network (DBN) [34], and a convolution neural network (CNN) [28]. By using the time, frequency, and time–frequency domain features, Deng et al. proposed a deep Boltzmann machine (DBM) for multiple fault classification of a rolling bearing with high reliability [35]. In [36], the combination of a sparse auto-encoder and a DBN is applied to classify faults by fusing the time- and frequency-domain features from multiple signals. The combined technique using a DBN and the Dempster–Shafer theory was employed to build an intelligent model for classifying the fault severities by Yu et al. [37]. In some cases, the CNN-based method performs better in comparison with the DBN method in complex scenario applications due to the advantage of sparse connection and weight-sharing ability [38].

Many researchers use CNNs with raw time-series signals received from the sensors for fault classification. Ince et al. [39] used a raw motor-current signal to design a 1-D CNN for real-time monitoring where the feature extraction and classification were combined in a single learning entity. In [40], the authors proposed a 1-D CNN for condition monitoring which automatically extracted features and performed better than a manual feature selection-based method. A CNN model with an adaptive learning rate and a momentum component was proposed in [41] for both fault-pattern identification and fault-size measurement with the raw signal. Another one-dimensional CNN with a deeper residual layer was designed by Peng et al. [42], where features were learned adaptively from time-series vibration signals and achieved quite a high accuracy on wheelset bearings.

Even though CNNs can learn distinctive patterns from raw data directly, in many cases, the raw data are highly corrupted by noise from the environment. Therefore, various domain-based processing algorithms incorporated with CNNs have been used to improve the fault-diagnosis system in recent times. Several methods exist to represent the original 1-D machinery data in a 2-D structure. Further, generating images from 1-D signals by applying different techniques, such as wavelet packet transforms (WPT), spectrograms, etc., and classifying them with a 2-D CNN model is found to be superior in the field of fault diagnosis [43,44]. After the initialization of the CNN, different modified architectures of CNNs were introduced, such as Res-net [45] and VGG-net [46], for improving feature extraction from images and achieving high accuracy. In [47], an S-transform was added with a CNN to construct a novel ST-CNN method, where the first S-transform converted the sensor data into a 2-D time–frequency domain matrix, and the CNN was later applied to perform fault classification. The envelope spectrum of an original signal using a Hilbert transform is an efficient method to convert the 1-D data into 2-D. Then, a deep CNN model was applied to learn the underlying features and classify different types of faults in [48].

Different types of wavelet transforms have been applied as data-to-image conversion techniques which mostly produce two-dimensional images and will later be used as the input of classifiers [43,49]. In [50], the Morlet function was applied in CWT to convert the time-series vibration signal to scalogram images. It was also used as the input of the convolutional attention neural network (CANN) to classify motor faults. In addition, the scalogram image-conversion technique also become a very efficient technique in the field of chatter identification and detection by creating distinctive images of every fault condition [51]. In [52], the image conversion was carried out with the bispectrum method, and a probabilistic neural network was proposed for image classification. Further, the authors showed that the converted 2-D grayscale images could produce different patterns among various fault types, and then various CNN models were applied to automatically extract features and classify them [53,54]. The above discussion represents the CNN model as a decent solution to diagnosis fault classification because it does not require expertise and it works in a noisy environment.

In this analysis, a novel signal-to-image conversion method named Gramian angular field (GAF) is applied to generate images [55,56]. The characteristics of the time-series signal are stored in polar coordinates and then converted into two distinct types of images by applying geometric operations. Finally, the generated images are classified with deep CNN. As a CNN is inherently an excellent choice for solving classification problems involving images, we aimed to convert 1-D time-series data to images through GAF encoding so that the CNN could be fed with the type of data it naturally excels at. Through this work, we prepared an equivalent GAF-image dataset from the KAT current signal and proposed a CNN model that was good at classifying bearing faults. Here, the GAF algorithm helps to increase the interpretability of switching from 1-D signals to 2-D images and provides a foundation for the effective extraction of features, and the CNN’s weight-sharing mechanism results in a much faster training speed than other networks. Our results indicate that such an encoded dataset (time-series to the image) could be a novel approach to the fault-classification research area.

The remainder of the paper is organized as follows: the detail of the theoretical background is provided in Section 2; the experimental testbed, data arrangement, and proposed methodology based on GAF and CNN are described in Section 3; the experimental results analysis and comparison with some build-in models and previous works are summarized in Section 4; and the concluding remarks are provided in Section 5.

## 2. Theoretical Background

### 2.1. Bearing Fault Frequencies

Among the various elements in IMs, the rolling element bearings (REB) are regarded as the most crucial elements due to their ability to reduce friction to operate the rotor smoothly. The bearings work as an electromechanical interface between the stator and the rotor, as well as a holding element to guarantee proper rotation from the shaft. Two types of races, named the inner and outer race, a group of rolling balls, and the cage where all the balls are enclosed with equal distance, are the basic elements of bearings. Faults in bearings can occur for multiple reasons, such as extreme load, wrong installation, misalignments in the rotors, inappropriate lubrication, and material fatigue [57]. The fundamental structure of the bearing and two fault conditions that are considered in this research (outer race fault and inner race fault) are shown in Figure 1.

In general, every bearing element has a fundamental rotating frequency. During rotation, if the rolling part passes through the damaged area of either the inner race or outer race, a periodic impulse is generated due to the rise of vibration energy at a fixed rate. This resultant frequency of the defect signal is known as the defect frequency, which can be estimated with the geometric parameters (diameter of the rolling element, cage, pitch, and the number of balls) and the rotation speed from the equations given in (1)–(3):(1)$$\mathrm{Outer}\;\mathrm{race}\;\mathrm{fault}\;\mathrm{frequency}:\;f_{O}=\frac{N_{ball}}{2}\times f_{m}\times \left(1-\left(\frac{D_{ball}}{D_{cage}}\times cos\beta \right)\right)$$
(2)$$\mathrm{Inner}\;\mathrm{race}\;\mathrm{fault}\;\mathrm{frequency}:\;f_{I}=\frac{N_{ball}}{2}\times f_{m}\times \left(1+\left(\frac{D_{ball}}{D_{cage}}\times cos\beta \right)\right)$$
(3)$$\mathrm{Ball}\;\mathrm{fault}\;\mathrm{frequency}:\;f_{b}=\frac{D_{cage}}{2D_{ball}}\times f_{m}\times \left(1-{\left(\frac{D_{ball}}{D_{cage}}\times cos\beta \right)}^{2}\right)$$

Here, $N_{ball}$ indicates the number of balls, $f_{m}$ is the rotational frequency, β is the load angle from the radial plane, and $D_{ball}$ and $D_{cage}$ represent the diameter of the ball and the cage, respectively.

As a result of bearing damage, a radial motion occurs between the stator and rotor which induces characteristic fault frequencies into the current signals and results in oscillations. Bearing defects cause radial displacements of the stator with the rotor, resulting in fluctuations in the load torque and the rotating eccentricity. Therefore, motor-current signals undergo amplitude, frequency, and phase modulation due to variations in machine inductances. The resultant motor-current signal $i\left(t\right)$ due to fault can be written as in Equation (4).
(4)$$i\left(t\right)=\sum_{k=1}^{\infty }i_{k}.cos\left(\omega_{c_{k}}.t+\phi \right)\;\mathrm{and}\;\omega_{c_{k}}=\frac{2\pi f_{bearing}}{p}.$$
where $\omega_{c_{k}}$ is the angular velocity, ϕ is the phase angle, and p is the pole number of the operating machine. The harmonic frequency due to the fault is denoted by $f_{bearing}=\left|f_{s}\pm mf_{v}\right|$. Here, $f_{s}$ and m denote the fundamental frequency and harmonic index number, respectively. $f_{v}$ will be either the inner race frequency ($f_{inner})$ or the outer race frequency ($f_{outer}$), depending on the fault that occurred.

### 2.2. Transformation of Time-Series Data into Images

The idea of transforming time-series data into 2-D images has evolved due to the rapid expansion of computer vision techniques. An efficient data transformation approach can reduce any massive amount of data into 2-dimensional or 3-dimensional feature sets. This equivalent visualization of time-series data provides a better understanding of the input data and helps to distinguish different types of signal conditions [56].

In our work, the transformation from time-series data to the image is divided into two stages. In the first step, data segmentation is applied to convert the long 1-D time-series data into multiple segments, and after that, the GAF image conversion technique is applied to the segments individually to generate the 2-D images.

#### 2.2.1. Data Segmentation

A simple preprocessing method is applied to convert 1-D original time-series current signal data into 2-D images by applying an adjustable sliding window mechanism [58] to extract useful features and reduce the computational time of the method. The recorded long current signal is divided into multiple segments by sliding a predefined window length. The resulting small segments help to accommodate a fair amount of training data for the training phase of the DL model. Along with that, this segmentation mechanism solves the issue of handling lengthy 1-D data to the designed model and finally, stacking the current signal together.

In this mechanism, the total number of samples $C_{Nt}$ can be defined as:(5)$$C_{Nt}=\left(\frac{C_{Lt}-C_{Lf}}{C_{Ls}}\right)+1$$
where, $C_{Lt}$, $C_{Lf}$, and $C_{Ls}$ denote the total length of the current signal, the length of one frame, and the step size, respectively. Here, the total length of the current signal is fixed, and the step size must be less than the frame length. Both the frame size and step size value should be set appropriately to generate enough samples. A larger frame size makes the input layer big, which results in high computational time in the neural network processing. On the other hand, a small frame may not cover the proper characteristics of the current signal and produce low classification accuracy. Similarly, a large step size generates fewer samples, which hampers the training method, and small steps result in many samples with similar characteristics. Hence, depending on the experiment and collected data size, the sizes must be defined appropriately. The overall process of the sliding window mechanism is illustrated in Figure 2.

#### 2.2.2. Gramian Angular Field (GAF)

The main idea of transforming time-series data into images with GAF is by implementing a matrix based on polar coordinates, which preserves the temporal correlation between the time-series signal of 1-D and the resultant Gramian matrices. Hence, using the polar coordinates helps to maintain absolute temporal relations in comparison with the Cartesian coordinates [59]. Two types of GAF images can be generated, namely, Gramian angular summation field (GASF) and Gramian angular differential field (GADF) by following the steps:(i)Normalization of the time-series data

In the first step, the input time-series signal is normalized within the minimum value of 0 and maximum value of 1. The normalization operation is defined as (6):(6)$$\tilde{b_{i}}=\frac{b_{i}-min\left(b\right)}{max\left(b\right)-min\left(b\right)}$$

Here, $\tilde{b_{i}}$ and $b_{i}$ represent the resultant signal after normalization of b and the raw current signal at time i, respectively;(ii)Transforming normalized data to polar coordinates

The second step of generating GAF images requires converting the normalized time-series signal to polar coordinates. This operation is performed by computing the angular cosine of every normalized value and time stamp as the radius. The formula of the polar coordinates can be expressed as:(7)$$\theta =arccos(\tilde{b_{i}});\;\mathrm{where}\;0\leq \tilde{b_{i}}\leq 1,\tilde{b_{i}}\in \tilde{B}\;r=\frac{t_{i}}{N};\;\;t_{i}\in N$$
where θ indicates the time-series value in the polar coordinates for every observation. $t_{i}$ and N are the time stamp and a constant stabilization factor for the space of the polar coordinate system, respectively. The range of the angle after applying this operation will be in [$0,\;\frac{\pi }{2}$]. The polar representation provides a better scenario for understanding the time-series data;(iii)Calculating GASF and GADF

Finally, by applying the trigonometric operation, the resultant polar coordinates of the original time-series signal can be converted into two different types of GAF. The trigonometric sum and difference will apply to sample points and the time correlation is generated from the angle perspective. The mathematical representations of GASF and GADF in matrix format can be written as Equations (8) and (9), respectively:(8)$$GASF=\left[\begin{array}{ccc} cos(\theta_{1}+\theta_{1}) & \cdots & cos(\theta_{1}+\theta_{n})\\ \vdots & \ddots & \vdots \\ cos(\theta_{n}+\theta_{1}) & \cdots & cos(\theta_{n}+\theta_{n}) \end{array}\right]$$
(9)$$GADF=\left[\begin{array}{ccc} sin(\theta_{1}-\theta_{1}) & \cdots & sin(\theta_{1}-\theta_{n})\\ \vdots & \ddots & \vdots \\ sin(\theta_{n}-\theta_{1}) & \cdots & sin(\theta_{n}-\theta_{n}) \end{array}\right]$$

This procedure of converting 2-D images with GAF can effectively maintain the order of the original time-series signal from the top left to the bottom right. For an input time series with a length of $n^{2}$, the resultant matrix dimension of GAN transformation will be n×n, where the original information is preserved in a positive diagonal and the relation between other time sequences is reflected in other regions of the matrix. Figure 3 depicts the steps of normalization, a transformation of the coordinate axis, and applying trigonometric functions that are involved in the process of converting 1-D time-series data to 2-D images using GAF.

### 2.3. CNN Model

The convolution neural network (CNN) was proposed by Le Cun et al. [60] as a branch of the neural network, developed for object recognition and based on functionalities of the human visual cortex. It is widely used in computer vision and image classification due to the similarity of the brain’s simple and complex cells of the visual cortex. This network is also considered an effective feed-forward supervised machine-learning network as it can be treated as the most efficient deep-learning method when large-scale CNN is considered. Generally, a CNN consists of a convolution layer (CL), a pooling layer (PL), and a fully connected (FC) layer. The overall structure performs feature extraction and classification, where finally the decision is expressed as a probabilistic function [61].

#### 2.3.1. Convolution Layer

The first layer of a CNN is named the convolution layer, which primarily extracts various input features from the input. Multiple rectangular neurons are added together to create a feature map. Neurons from the same map generally share weights, which are known as convolution kernels. These kernels are initialized in the form of random matrices. The sharing weights in the convolution layer help to reduce the possibility of overfitting by minimizing the connection among the layers of the network [62].

Several learnable kernels are implemented in this layer, and a convolution operation is performed between these kernels and the input image to generate feature maps. Then, these maps are used as inputs to the activation functions to implement a nonlinear operation.

The output of the convolution operation of each layer can be mathematically calculated as follows:(10)$$x_{j}^{l}=f\left(\underset{i=1,2,\ldots ,M}{\sum }x_{i}^{l-1}\times k_{ij}^{l}+b_{j}^{l}\right),j=1,\ldots ,N$$
where $x_{j}^{l}$ denotes the *j*-th output map of the convolution layer for the filter, *k*, and the number of inputs, *M*. Further, $x_{j}^{l-1}$ represents the i-th input feature map of (*l* − 1) layer, $b_{j}^{l}$ implies the bias value *j*-th filter, and f denotes the activation function. The number of layers of a CNN completely depends on the variation, as well as the complex pattern of the input. If the feature variant is quite high, a deep CNN must be designed to diagnose them all accurately. ReLU is the most used activation function in CNNs for increasing nonlinearity, which can be defined as $x_{i}^{j}=max(0,x_{i}^{j^{,}})$.

#### 2.3.2. Pooling Layer

To reduce the computational time, the pooling layer is connected after the convolution layer, which reduces the feature map size with a down-sampling operation without changing the variance of the distinguishing feature scale. The connected pooling layer merges the analogous features in a local position without altering the exact features and finally, the produced output is less sensitive to the surroundings. The output feature maps of the *l*-th layer can be calculated as:(11)$$x_{j}^{l}=f\left(\beta_{j}^{l}.down\left(x_{j}^{l-1}\right)+b_{j}^{l}\right),j=1,\ldots ,M$$

Here, $x_{j}^{l}$ and $x_{j}^{l-1}$ are the *j*-th output and input map. *f* and *down(.)* denote the activation function and sub-sampling function, respectively. Two bias operations, multiplicative bias and additive bias for the *j*-th filter, are denoted by $\beta_{j}^{l}$ and $b_{j}^{l}$.

#### 2.3.3. Fully Connected Layer

The final one or more fully connected layers are connected after adding multiple convolutions and pooling layers to evaluate the results from previous layers and classify the output. For the input length, *M*, and total output vector length, *N*, the resultant output of the *l*-th layer can be calculated as follows:(12)$$x_{j}^{l}=f\left(\underset{i=1,2,\ldots ,M}{\sum }x_{i}^{l-1}\times w_{ij}^{l}+b_{j}^{l}\right),j=1,\ldots ,N$$
where $x_{j}^{l}$ denotes the *j*-th output value, $x_{j}^{l-1}$ is the *j*-th input value, $b_{j}^{l}$ and $w_{ij}^{l}$ are the bias and weight of the *j*-th output, and f indicates the activation function for the fully connected layer. In general, for the classification problem, the output of the fully connected layer is a probability for every class or category, computed by a SoftMax activation function.

The CNN model that is applied includes two convolution layers with filter sizes of 16–3 × 3 and 32–3 × 3, where the input image size is 128 × 128 × 3. The max-pooling layer has a size of 2 × 2. Finally, a fully connected layer is added to classify three different health states of the IM. The structure of the deep CNN model that is used in this study is provided in Figure 4.

The detailed sequential model of the designed deep CNN is provided in Table 1, which includes the types of layers, activations in each layer, and the number of parameters of each layer.

The trainable parameters are first initialized in the CNN model, followed by optimization using the adaptive second estimation (Adam) technique to reduce the error between the original and predicted values. To measure the degree of training error, categorical cross-entropy is used. Several types of CNN models were evaluated by varying each of the defined layers, and it was discovered that the large deep model of CNN not only provided higher accuracy, but also took a long time to train. In the training phase of the deep-learning model, the suggested CNN runs for 50 epochs. During the experiment, the dataset was randomly split into training, validation, and testing subsets and repeated 30 times to eliminate any possibility of contingency during the test, and the average values of the experiments were considered as the final outcomes for analysis. Additionally, three other deep neural network architectures, GoogleNet, ResNet-18, and AlexNet, were also trained with identical 2-D images for comparison with the proposed model’s findings.

## 3. Methodology

### 3.1. Experimental Testbed

The current signal from the Paderborn University bearing dataset was used in this work, which was developed by the mechanical engineering research center of Kat-Data Center and described in [63]. In this massive dataset, a wide range of bearing conditions have been considered; the dataset contains data from healthy bearings, data for artificially damaged bearings, and finally, real damage data that are generated by an accelerated lifetime test. For creating artificial damage on the bearing, drilling, electric discharge machining (EDM), and manual electric-sculpting methods are used. In the case of the accelerated lifetime test, two different damage methods, named fatigue (F) and plastic deformation (P), were applied. The test rig followed the ISO 15243(2010) standard in terms of the damage size, geometry, location, and type of occurrence. Different types of data, such as vibration, current, speed, and temperature were recorded for every bearing condition. In our work, we used the current signals of two different phases. The testbed contains a modular system consisting of an electric motor, a torque measurement shaft, a test module of a rolling bearing, a flywheel, and a load motor, as demonstrated in Figure 5. Here, a frequency inverter with a 16 kHz switching frequency was used to control a 425 W permanent magnet synchronous motor (PMSM). A LEM CKSR 15-NP modelled current transducer measured the current signal in two different phases and a 25 kHz low-pass filter filtered the measured signal and converted it to a digital signal with a sampling rate of 64 kHz.

The experimental dataset contains 32 different bearings with various degrees of damage. In our analysis, we considered current signals from 17 bearings corresponding to the healthy condition, outer ring damage (ORD), and inner ring damage (IRD). These three states are labelled as class 0, class 1, and class 2, respectively. Table 2 summarizes the details of the bearing conditions that are considered in this work. Each bearing was run 20 times under every load condition. In every case, the current signal was recorded for 4 s with a sampling rate of 64 kHz. Thus, each recording has 256,000 sample points (approximately).

Four different operating load conditions were considered during data collection, as provided in Table 3. We consider the data for 1 s containing 64,000 points for every condition and after that divide them into multiple segments to generate enough images for classification with 2D-CNN. Depending on the operating conditions, we split the total dataset into 4 segments, and every part contained three described bearing health conditions.

### 3.2. Proposed Method

Figure 6 illustrates the overall methodology of the bearing fault classification using the motor-current signal.

According to this figure, the proposed methodology of the fault-diagnosis approach can be divided into the following steps:(i)Split data based on operating conditions

As discussed in Section 3.1, we consider the motor-current signal for this study, which includes four different working conditions (Table 3). Firstly, we split the overall data into four sets based on these conditions. The segmented data consist of three different health conditions of the bearing (healthy, outer, and inner). Instead of considering 4 s of signal, we considered 1 s of the current signal.(ii)Data segmentation

The raw current signal needs proper segmentation to make it appropriate for the transformation by using the encoding schemes. We applied a sliding-window technique, which is discussed in Section 2.2.1, and created multiple signals from a long signal that preserved the properties of the original signal.(iii)Transforming to 2-D images

After proper segmentation of the original current data, the GAF algorithm (Section 2.2.2) is applied to each segment to convert the 1-D signal into 2-D images. Here, the time-series data are converted to a polar coordinate system before being converted into an image, which helps to project the time-series amplitude variation to the angular variation in the polar coordinate. This projection helps to create significant variation among the generated images due to different types of bearing conditions.(iv)Classification using a CNN

Finally, the generated image dataset is used as the input of the deep CNN model (described in Section 2.3) for training and testing. Then, we compare the performance of our designed model with some other existing models, as well as some previous works, to validate our proposed fault-diagnosis model with the current signal data.

### 3.3. Performance Evaluation Parameters

In any classification problem, the samples can be classified into four different categories, depending on the original class and the predicted class output. The categories are: TP (True Positive: both the original and predicted class are positive); FP  (False positive: the original class is negative, but the predicted class is positive); FN (False negative: the original class is positive, but the predicted class is negative); and finally, TN (True Negative: both the original and predicted class are negative). By using these categories, we evaluate the model performance with four commonly used matrixes named recall, precision, F1-score, and model accuracy (Equations (13)–(16)).
(13)$$Precision\;\left(P\right)=\frac{TP}{TP+FP}$$
(14)$$Recall\;\left(R\right)=\frac{TP}{TP+FN}$$
(15)$$F1\_score\;\left(F1\right)=\frac{2\times Precision\times Recall}{Precision+Recall}$$
(16)$$Accuracy\;\left(Acc\right)=\frac{TP+TN}{TP+FP+TN+FN}$$

Additionally, a visual summary of the classification results can be represented with the four mentioned categories, which is known as the confusion matrix. From this matrix, the correctly and incorrectly predicted values of the model can be expressed with numerical values and the misclassified samples can be easily understood.

## 4. Experimental Results

### 4.1. Analysis of Current Signal Imaging with GASF and GADF

The motor-current signals that are recorded from the Paderborn University bearing dataset are considered to generate the 2-D image by applying the Gramian angular field (GAF) image encoding technique. As we mentioned earlier, the overall dataset is split up based on the four working conditions, where each condition contains three health states of the bearing (normal, outer race fault, and inner race fault), which helps to create distinct patterns for each working condition. To produce enough images for the classification with the CNN, we segmented the 1 s of the current signal data into multiple portions before encoding them as images. Figure 7 presents the resultant images after applying the GASF and GADF for the four different working conditions in three different states of the bearing.

A total number of 3600 images are created from current signal segments that represent all the operating conditions. Among them, 2304 images are used for training and 576 images are used for validation. After completing the training phase, the remaining 720 images are used as the testing samples to evaluate the model performance. The sample splitting ratios for every working condition are presented in Table 4.

### 4.2. Diagnosis Performance of the Proposed Method

Our proposed CNN model (detail architecture in Table 1) containing two convolution layers took the resultant images as the input to classify the bearing conditions. Transforming the time-series data to polar coordinates with GAF helps to create individual patterns for each considering condition, which helps the CNN model to automatically learn and extract features, and finally, classify the normal or faulty bearing conditions successfully. The results are provided in Table 5. The resultant images for the four working conditions mentioned in Table 4 are individually grouped for both GASF and GADF. The images for GASF are named GASF_1, GASF_2, GASF_3, and GASF_4, and the images for GADF are labelled as GADF_1, GADF_2, GADF_3, and GADF_4 for the four different conditioned data, respectively. It can be observed that the proposed approach can achieve excellent performance, attaining test accuracies of approximately 99% to 100% for all four working conditions with GADF- and GASF-encoded images. Not only the accuracy, but also the other three performance matrices show similar scores. The experiments were repeated 30 times to test the repeatability of the model, and in every case, the values of the accuracy matrices were between 99% and 100%. Therefore, to validate the performance of the proposed model, three different popular existing CNN models, GoogleNet, ResNet-18, and AlexNet, were also trained and tested with the generated 2-D images. The evaluation parameters presented in Figure 8 indicate that the benchmark CNN models also provide high classification performance with GADF and GASF images, which implies that this imaging approach can be coupled with a CNN for bearing fault classification from the current signal. Moreover, with our proposed model, it is evident that a very good classification accuracy can be achieved using CNN architecture with relatively small depth when GADF and GASF images are used. Therefore, computation complexity and training time are also reduced in comparison with the existing CNN architectures that are mentioned above.

Finally, we merge all the generated images for every working condition and train the CNN model to classify the bearing health conditions. The accuracy of this model is 99.58%, which also demonstrates that the generated images with GAF can successfully create distinguishable patterns for every condition and help the CNN model to achieve high classification accuracy in classifying bearing conditions. All the presented results are generated with the train-to-test ratio of 80:20. Along with this, we varied the train and test ratio to observe whether or not the designed CNN model could train the model with a smaller number of samples. The accuracy of the model with different train and test image sets is provided in Table 6. It is shown that for a low amount of training samples, the performance is quite high with the designed model.

Figure 9 represents the training and validation accuracy and loss curves of the deep CNN model for 50 epochs. The classification accuracy of the training phase reached 99.58%, which is also true for the validation dataset.

A feature mapping technique was applied to verify the self-learning ability of the designed deep-learning method using t-distributed stochastic neighbor embeddings (tSNE). As shown in Figure 10, different bearing conditions of motor-current signals were easier to recognize using the deep CNN model. The visualization of each layer of the CNN demonstrates how the nonlinear mapping helps to capture prominent features in every step. In the early layers, none of the fault signals were separable, and the model could not perform well. However, deeper in the layers, the system can fully utilize the self-learning capability for fault classification. As a result, three bearing conditions of the current signals are properly clustered in the final layer.

### 4.3. Comparison with Some State-of-the-Art Methods

We compare the outcomes of our proposed method with some basic ML techniques in the field of fault diagnosis by validating its performance. In the first approach, the CNN model was kept unchanged, and the original segmented current signal as mentioned earlier was used as the input to see if the transformation technique from 1-D to 2-D signal contributed significantly to model performance. This approach [64] (Original + 1-D CNN) could not perform well and only achieved a 61.67% accuracy. Later, the continuous wavelet transform (CWT) techniques were applied to the original current signal and converted to 2-D images. They were then classified with the same designed CNN model (CWT + 2-D CNN) [65]. Here, the resultant images fail to generate a significant pattern in different bearing conditions and exhibit a classification accuracy of 64.58%. Due to the drawbacks of the recorded current signal, such as low signal-to-noise ratio (SNR), saturation harmonics, and information loss, the discussed techniques cannot attain superior performance in this scenario. Both models showed significantly lower accuracy, precision, recall, and f1_score values in comparison to our proposed method, which is shown in Table 7.

The resultant confusion matrix of the three above-mentioned methods is provided in Figure 11a–c, where only data from a single operating condition were considered. Finally, Figure 11d represents the confusion matrix, where data for all four conditions were taken into account to observe the wrongly detected samples of each bearing condition with our proposed fault-classification approach.

### 4.4. Comparison with Existing Works

As a final step, we compared our bearing fault classification approach with some other existing methods that utilize the same dataset of motor-current signals as listed in Table 8. Lessmeier et al. [63] used a wavelet packet decomposition method up to three levels, along with a special SVM approach called SVM-PSO (SVM-particle swarm optimization). This method achieved an accuracy of 86.03%. In [61], the combination of an information fusion and grayscale image conversion technique was applied to convert the time-series signal, and finally, the images were classified with three supervised algorithms (Multilayer perceptron, Support vector machine, and k Nearest Neighbor). Therefore, with the same dataset of the current signal, Hsueh et al. used the empirical wavelet transform to transform the 1-D current signal into 2-D grayscale images, and later to classify the fault using a CNN with 97.3 (%) accuracy [27].

The GAF-based imaging technique is used in various fault-classification scenarios. In this work, we also demonstrated that this technique could exhibit superior performance for motor-bearing fault classification when the current signal is considered. A brief comparison of several other scenarios is tabulated in Table 9. Although the aims of these studies may be different, they employed the GAF-based imaging method, and the research outcome proved it to be an effective technique to convert time-series data into 2-D images.

## 5. Conclusions

Intending to improve both the diagnosis capability and adaptability of the approach for diagnosing faults in an IM’s bearing system under different working conditions, we presented a new technique that was based on GAF image encoding and a deep CNN model. Data-driven fault-classification approaches became one of the focused research areas in the fault-diagnosis field because of the availability of a massive amount of sensor data. Due to the low cost, easy accessibility, and smooth data acquisition capability of the motor-current signal, it is considered to be a smart solution among the various available sensor data. The proposed method incorporates a GAF-based image generation technique from a 1-D current signal for a 2-layer CNN model to construct a data-driven intelligent fault-diagnosis approach for bearings. Firstly, the original current signal data are divided based on the working conditions and then segmented into multiple samples for image conversion. Two different types of images named GASF and GADF were generated where the time-series information was converted into polar coordinates. As the current signal is affected by surrounding noises, it becomes very difficult to extract the fault signatures manually. When the data are converted into the polar coordinates for image transformation, the different bearing-condition data create distinctive patterns, which helps the CNN model to easily extract the necessary high-level features. In all considering operating conditions, this proposed GAF + 2-D CNN based approach can attain good accuracy. Three predefined CNN models, GoogleNet, ResNet-18, and AlexNet, were also trained and tested with the GAF images to validate the image encoding technique in the fault classification area. The shallower depth of our proposed CNN model also results in less training time and computation complexity. Moreover, a comparative study was conducted with some reference approaches, as well as with recent works, to examine the effectiveness of the proposed approach. According to the analysis of all experimental results, our overall method achieves more than 99% accuracy, and the same is true for the other performance measuring parameters. However, in this work, we considered only one dataset, as few current signal datasets for bearing faults are publicly available. In the future, we would like to test the performance of our proposed methodology in other datasets. Further, we only considered the presence of a fault in the bearings when the data were preprocessed, but fault severity was not considered. In the future, we also intend to analyze the performance of this proposed model using a fault severity analysis. To make the decision-making process more automated and enhance the reliability and robustness of the system in future research, we plan to extend the proposed technique with multiple sensors and integrate a systematic hyperparameter tuning approach for the CNN structure.

## Figures and Tables

**Figure 1 sensors-22-04881-f001:**
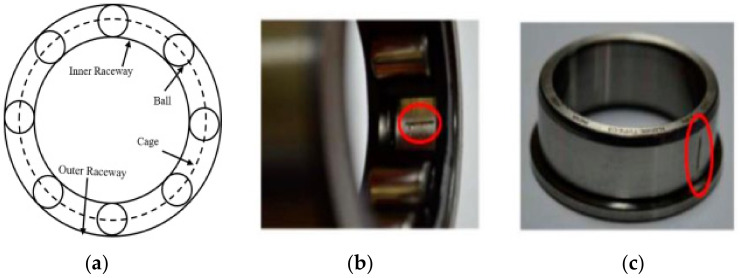
(**a**) Bearing geometry; (**b**) fault in the outer race; (**c**) fault in the inner race.

**Figure 2 sensors-22-04881-f002:**
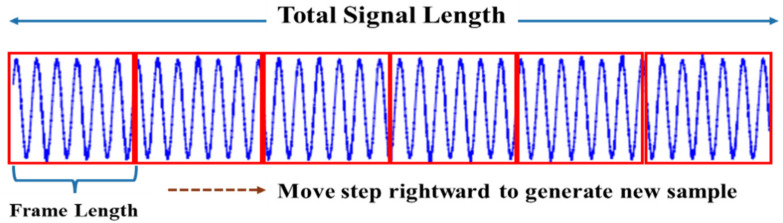
Data segmentation for converting time-series data into an image.

**Figure 3 sensors-22-04881-f003:**
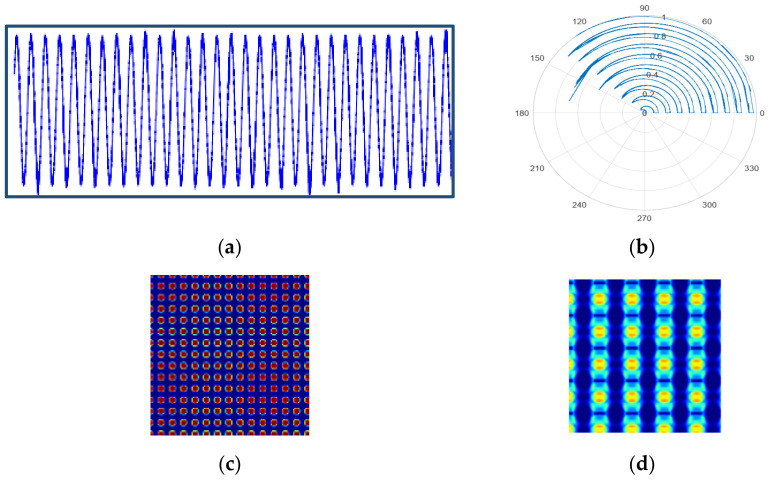
Steps of GAF: (**a**) normalized time-series signal; (**b**) converted signal in polar coordinates; (**c**) GASF; and (**d**) GADF.

**Figure 4 sensors-22-04881-f004:**
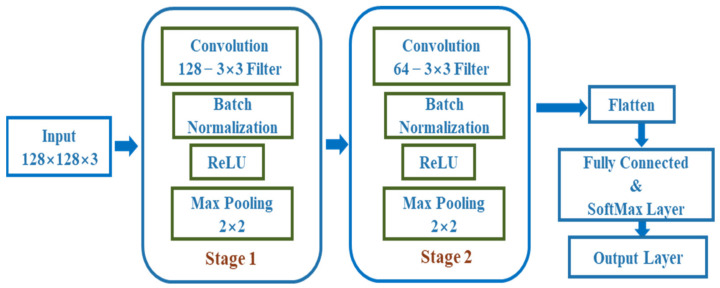
The modified architecture of the deep convolution neural network.

**Figure 5 sensors-22-04881-f005:**
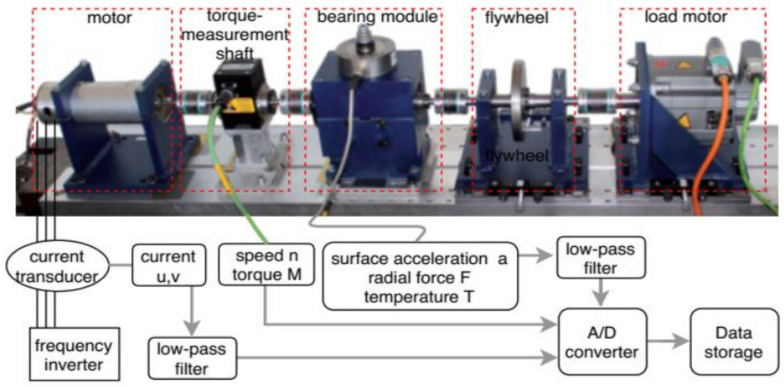
Organization of the UPB bearing test rig.

**Figure 6 sensors-22-04881-f006:**
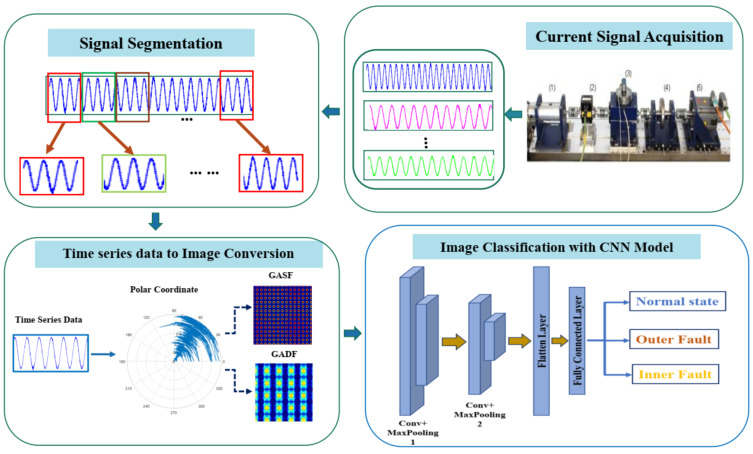
The workflow of the proposed method.

**Figure 7 sensors-22-04881-f007:**
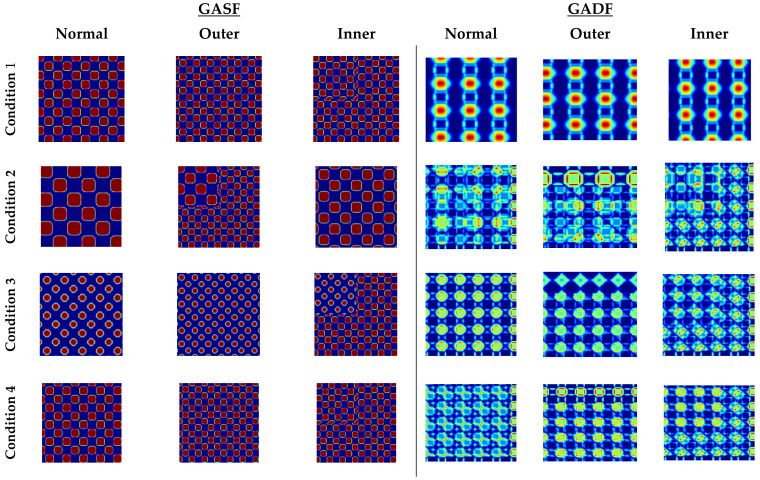
The resultant 2-D images after applying GASF and GADF algorithms on four considering working conditions.

**Figure 8 sensors-22-04881-f008:**
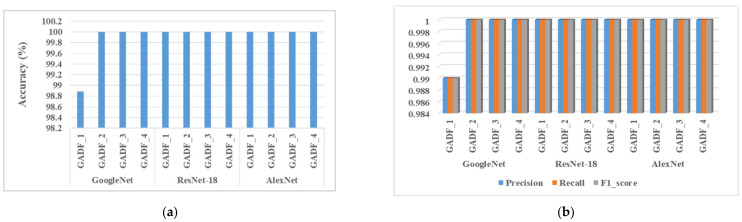
The performance of the three existing models: (**a**) accuracy and (**b**) precision, recall, and F1_score for the four conditioned GADF-encoded images.

**Figure 9 sensors-22-04881-f009:**
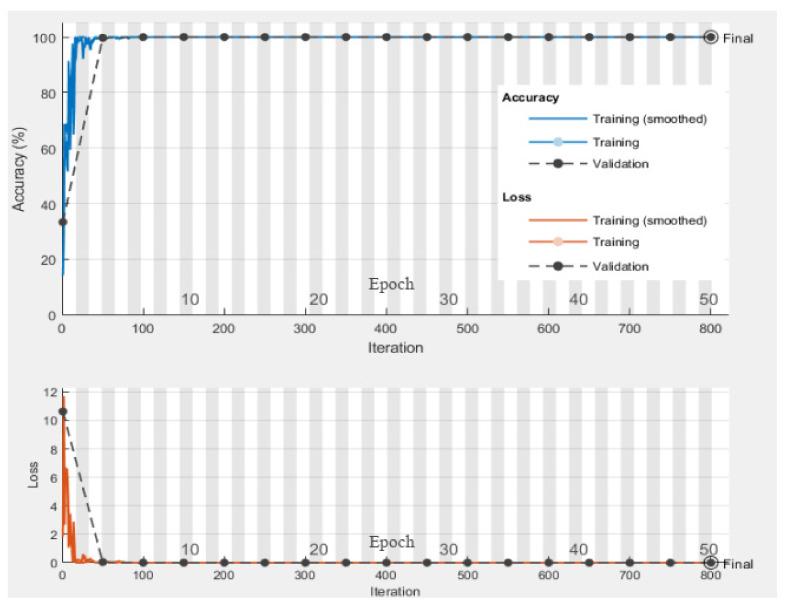
Accuracy and loss curve of the deep CNN model.

**Figure 10 sensors-22-04881-f010:**
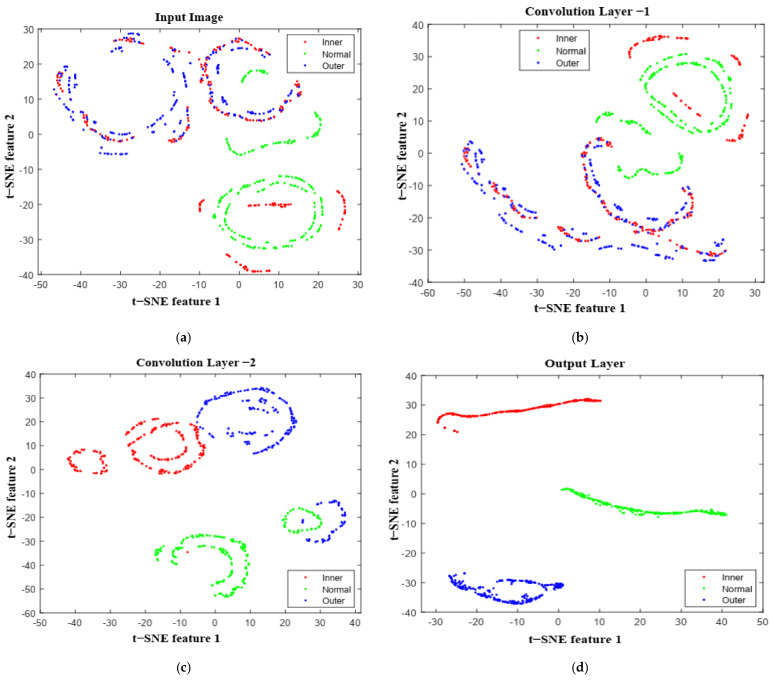
Feature visualization via t−SNE: (**a**) input image; (**b**) initial convolution layer; (**c**) final convolution layer; and (**d**) output layer.

**Figure 11 sensors-22-04881-f011:**
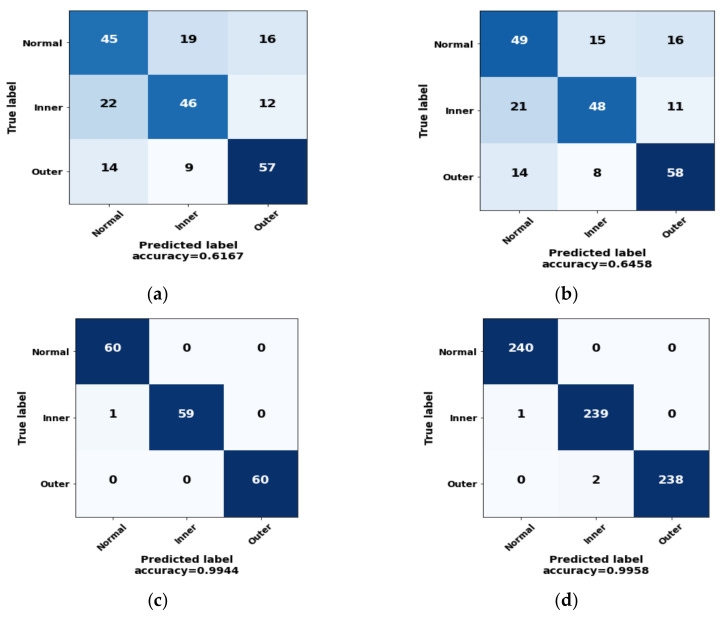
The confusion matrix of three different basic techniques: (**a**) Original + 1-D CNN; (**b**) CWT + 2-D CNN; and (**c**) GAF + 2-D CNN for a single condition data. (**d**) GAF + 2-D CNN for the complete dataset.

**Table 1 sensors-22-04881-t001:** Layer-wise details of the deep CNN.

Layer (Type)	Activations	Number of Parameters
conv1d_1 (Conv1D)	128 × 128 × 16	448
Batch_Norm1 (Batch Normalization 1)	128 × 128 × 16	32
ReLU_1	128 × 128 × 16	0
max_pooling1d_1 (MaxPooling)	64 × 64 × 16	0
conv1d_2 (Conv1D)	64 × 64 × 32	4640
Batch_Norm2 (Batch Normalization 1)	64 × 64 × 32	64
ReLU_2	64 × 64 × 32	0
max_pooling1d_2 (MaxPooling)	32 × 33 × 32	0
FC (Fully Connected)	1 × 1 × 3	101,379
SoftMax	1 × 1 × 3	0
Output Class	-	0
Total params: 106,563		
Trainable params: 106,563		
Non-trainable params: 0		

**Table 2 sensors-22-04881-t002:** Characterization of bearings considered in this work.

Type of Bearing		Bearing Code	Damage Extent	Damage Method	Label
Healthy bearing (H)		K001	-	-	0
		K002	-	-	
		K003	-	-	
		K004	-	-	
		K005	-	-	
		K006	-	-	
Naturally damaged Bearing	Outer ring damage (ORD)	KA04	1	F	1
		KA15	1	P	
		KA16	2	F	
		KA22	1	F	
		KA30	1	P	
	Inner ring damage (IRD)	KI04	1	F	2
		KI14	1	F	
		KI16	3	F	
		KI17	1	F	
		KI18	2	F	
		KI21	1	F	
F = fatigue: pitting; P = Plastic deform: indentations.					

**Table 3 sensors-22-04881-t003:** Details of the four different operating conditions of rolling element bearings.

Operating Conditions	Rotational Speed (S) [rpm]	Load Torque (M) [Nm]	Radial Force (F) [N]	Bearing Heath Type
**Condition 1**	1500	0.7	1000	H/ORD/IRD
**Condition 2**	900	0.7	1000	H/ORD/IRD
**Condition 3**	1500	0.1	1000	H/ORD/IRD
**Condition 4**	1500	0.7	400	H/ORD/IRD

**Table 4 sensors-22-04881-t004:** The splitting ratio of the dataset into training, validation, and test sets.

	Training (80%)		Testing (20%)	Sample Count	Sample/ Condition
Dataset	Training (80%)	Validation (20%)			
Condition 1	576	144	180	900	300
Condition 2	576	144	180	900	300
Condition 3	576	144	180	900	300
Condition 4	576	144	180	900	300
	2304	576	720		

**Table 5 sensors-22-04881-t005:** The performance measurement of the designed CNN architecture.

Datasets	Accuracy (%)	Precision (P)	Recall (R)	f1_Score (f1)
GADF_1	99.44	0.99	0.99	0.99
GADF_2	100	1.0	1.0	1.0
GADF_3	100	1.0	1.0	1.0
GADF_4	98.89	0.98	0.98	0.98
GASF_1	100	1.0	1.0	1.0
GASF_2	100	1.0	1.0	1.0
GASF_3	100	1.0	1.0	1.0
GASF_4	100	1.0	1.0	1.0
Average	99.79	0.996	0.996	0.996

**Table 6 sensors-22-04881-t006:** Accuracy value for different train-test ratio.

Train-Test Ratio	Accuracy (%)
80:20	99.58
70:30	99.91
60:40	100
50:50	99.94
40:60	99.95
30:70	99.96
20:80	99.69

**Table 7 sensors-22-04881-t007:** The resultant evaluation matrices for three different approaches.

Techniques	Evaluation Parameters			
	Precision	Recall	f1_Score	Accuracy (%)
**Original + 1-D CNN**	0.61	0.59	0.61	61.67
**CWT+2-D CNN**	0.61	0.61	0.61	64.58
**GAF+2-D CNN** **(Proposed)**	0.99	0.99	0.99	99.44

**Table 8 sensors-22-04881-t008:** The classification results of some existing works.

Applied Models	Classification Accuracy (%)
WPD + SVM-PSO [63]	86.03
Information fusion + MLP [61]	98.0
Information fusion + SVM [61]	98.3
Information fusion + kNN [61]	97.7
EWT+CNN [27]	97.3
GAF+2-D CNN (proposed)	99.58

**Table 9 sensors-22-04881-t009:** Summary of GAF-based imaging techniques in different applications.

Serial No.	Ref	Aim of the Research	Methods Applied	Result	Dataset
1	[66]	Fault classification with vibration data	A lightweight CNN bearing fault intelligent diagnosis model combining GAF and coordinated attention (CA) (GAF-CA-CNN)	Accuracy: 99.62% Standard deviation: 0.154	Case Western Reserve University (CWRU) bearing vibration dataset
2	[67]	Classify time-series data	Time-series data to 2D images with GAF/MTF + Tiled CNN	Mean square error (MSE): 0.00889 with GASF images	ECG, CBF, Gunpoint, SwedishLeaf, and 7 Misc
3	[55]	Fault diagnosis and classification with vibration data	GAF and MTF techniques with capsule networks (GAFMTF-CapsNet)	Accuracy: 99.81%	CWRU bearing dataset
4	[56]	Predictive maintenance framework of conveyor motors	Principal component analysis (PCA) + GAF + CNN (used PReLU activation function)	Accuracy: 100%	
5	[68]	Sensor classification	Piecewise aggregate approximation (PAA) + GDF/MTF + ConvNet	Error rate: 0.4 (Wafer dataset) 5.35 (ECG dataset)	The Wafer and ECG databases
6	[69]	Human activity recognition classification	GAF + multi-dilated kernel residual network (Fusion Mdk-ResNet)	Accuracy: 97.27%	WISDM dataset, UCI HAR dataset, and Opportunity Dataset
7	[70]	Classification of the conventional faults in hydraulic component	An improved data-enhanced Gramian angular sum field (DE-GASF) + multichannel dual attention convolutional neural network (MC-DA-CNN)	Accuracy: 96.48% (axial piston pump fault) and 98.08% (hydraulic reversing valve fault)	
8	[71]	Bearing fault diagnosis with time-series vibration data	Piecewise aggregation approximation (PAA) with GAF + convolutional channel attention residual network (CCARN)	Accuracy: 100%	
9	[72]	An FDGAF-based intelligent wheel flat diagnosis technique	Frequency-domain Gramian angular field (FDGAF) + transfer learning network	Disparities between intra-class and inter-class distance for FDGAF under all four considered velocities	
10	This work	Bearing fault classification with motor-current signal	Image segmentation+ GAF + 2-D CNN	Accuracy: 99.58%	KAT bearing dataset (current signal)

## Data Availability

The data are publicly available.

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
