# Peer review of "A Bearing Fault Classification Framework Based on Image Encoding Techniques and a Convolutional Neural Network under Different Operating Conditions"

_sensors, 2022, doi:10.3390/s22134881_

Round 1

Reviewer 1 Report

The manuscript addresses a current research problem related to data-driven diagnostics of induction motors. The authors proposed a CNN-based bearing fault classification method, in which the recorded time series for the stator electrical currents were converted into 2D images using Gramian angular field.  Overall, the paper is well organized, well written, and an extensive literature review of relevant work in this area was provided. Although only a few English grammar mistakes were found in the manuscript, I suggest that the authors submit the manuscript through a thorough grammar review, preferably with a native English speaker. Despite the superior performance results obtained with the method proposed by the authors, I wonder if the method would also provide satisfactory results for electrical machines operating under more realistic conditions (for instance, with stochastic variation of loading conditions). It seems that the majority of the publicly available condition monitoring datasets consider data extracted from machines operating under well-defined steady conditions, however, the ability of such data-driven fault identification algorithms to effectively perform under stochastic operating conditions of electrical machines is what defines its real potential for realistic field applications.      

Reviewer 2 Report

This paper introduced a fault classification model that combines a signal-to-image encoding technique and a convolution neural network with the motor current signal to classify bearing faults. Overall, the paper is well organized, my comments are as follows,

1.                  The authors proposed a Gramian Angular Field (GAF) algorithm based convolutional neural network (CNN) method for bearing fault diagnosis. This idea is not new, there are many similar papers in recent years. Moreover, the paper does not present these methods in comparative studies.

2.                  The CNN model is quite simple, did the authors have any contribution to improving the learning model?, how to choose the optimal set of model parameters?. The authors should also showcase the results of the training and validation of the training model to see how it works.

3.                  The authors could further enlarge the Introduction with current work of recent state-of-art techniques for motor fault detection based on time-series signals and 2D-image to improve the research background, for example; Effective Fault Diagnosis Based on Wavelet and Convolutional Attention Neural Network for Induction Motors; Fusion of vibration and current signatures for the fault diagnosis of induction machines; Experimental setup for online fault diagnosis of induction machines via promising IoT and machine learning: Towards industry 4.0 empowerment; Milling chatter detection using scalogram and deep convolutional neural network.

4.                  The experiment setup and the data collection were not described clearly. How many data points/sample are chosen? Is any related to the sampling rate?

5.                  The authors stated that there are 576 images used for validation and 720 images used for testing. But in the confusion matrix figure, there are only a few samples on that, please explain about that?

6.                  How about the computational time of the proposed algorithm, please compared it to other methods.

7.                  In Figure 9: Feature visualization via t‐SNE. It is very clear to see the three classes already isolated from the input layer, therefore, they can be successfully classified with any classifiers.

8.                  How did the authors obtain the classification results of some existing works in Table 8? Have the authors used other architectures to run on the same dataset? Please clarify this point, including the model parameter setting for all models.

Round 2

Reviewer 2 Report

The manuscript has improved in the revised version. However, there are a few concerns:

1. Figure 9 should be revised. On the horizontal axis, both the number of iterations and epochs are presented but no caption for the number of epochs.

2. Fig. 11 (d) GAF 573 + 2-D CNN for the complete dataset was added but no further description about that. Differences between Fig 11.c and 11.d should be discussed.
